# Mezcal Characterization Through Sensory and Volatile Analyses

**DOI:** 10.3390/foods14030402

**Published:** 2025-01-26

**Authors:** Oxana Lazo, Ana Lidia García-Ortíz, Joaliné Pardo, Luis Guerrero

**Affiliations:** 1Centro de Investigación en Biotecnología Aplicada, Instituto Politécnico Nacional, Carretera Estatal Santa Inés Texcuexcomac Km 1.5, Tepetitla 90700, Tlaxcala, Mexico; olazoz@ipn.mx (O.L.);; 2Facultad de Derecho Acapulco, Universidad Autónoma de Guerrero, Paseo de la Cañada, Alta Progreso, Acapulco de Juárez 39610, Guerrero, Mexico; joalinepardo@uagro.mx; 3Food Quality and Technology, Institut de Recerca i Tecnologia Agroalimentàries—IRTA, Finca Camps i Armet s/n, Monells, E-17121 Girona, Spain

**Keywords:** mezcal sensory wheel, sensory panel training, Mexico mezcal, volatile analysis, distillation process

## Abstract

Mezcal is a traditional beverage with relevant cultural and economic importance in Mexico, with different Protected Designation of Origin locations. This study focuses on creating a sensory lexicon for Mezcal with local producers by means of Free Choice Profiling. A selection of the most relevant descriptors was made to construct a sensory wheel. Subsequently, a sensory panel evaluated a total of 10 Mezcal samples using the sensory categories defined in the sensory wheel. Additionally, gas chromatography with mass spectrometry was performed to analyze volatile components’ contribution to the aroma and flavor descriptors. A total of 87 terms were selected for the sensory wheel, using 41 descriptors within 10 categories for odor modality and 46 more within 13 categories for flavor modality. The main volatile compounds that were identified were 37 esters, 17 alcohols, 12 ketals and 9 terpenes, which were the foremost contributors to the presence of several sensory descriptors and were also found in most of the Mezcal samples. The quantitative analysis results exhibited a higher floral odor for Mezcal of the Angustifolia variety and the highest smoked odor for an earthenware distilled Mezcal, thus proving that the selection of the descriptors from the wheel was appropriate for differentiating Mezcal samples from different origins, agave species and distillation processes. Therefore, the sensory wheel developed in this study can be used both as a quality control tool and as a marketing tool that allows producers to differentiate their products in the market.

## 1. Introduction

Mezcal is a traditional beverage with relevant cultural and economic importance in Mexico. Its consumption has grown significantly in recent years [1], especially since it acquired the Protected Designation of Origin status in different Mexican regions, which recognizes its sensory identity linked to different production sites and elaboration processes. According to Gomis-Bellmunt et al. [2], most consumers have a positive attitude toward products with quality labels, such as the Protected Designation of Origin (PDO) label. In this vein, Mezcal production has increased to 147% since 2015 due to its growing number of producers, its diversity and its recent diffusion. Consequently, this product has been placed in international markets, including the European one [3].

Mezcal production starts with the manual harvesting of agave to obtain their hearts or pines; the leaves can also be recovered. The hearts are used whole or cut in half (according to size) and are cooked in a hole or oven in the ground that has been heated by burning wood (normally oak) to heat stones that retain heat. Once this oven is heated, agave hearts and sometimes leaves are placed inside and covered with grass, mud and stones to prevent the heat from escaping. Once cooking has finished, the agave hearts are removed and mashed, either manually with hammers and hatchets or by using a millstone. Mezcal fermentation is carried out using the whole mash from the agave heart, including the fibers. The fermentation vessel can vary depending on the producer and is made of either wood or leather, or some other material adapted for this process. Once fermentation is finished, the mash is transferred to a small scale (sometimes handmade) made of copper for the first distillation process. The second distillation process (known as rectification) is also carried out in the copper pot, and the spirit obtained should be approximately 50% ethanol (*v*/*v*). Furthermore, a chemical transformation of volatiles may take place when copper is used. Distillation is also relevant to the product’s flavor as it is in this step that the recovery of volatiles takes place by means of selective separation [4].

Mezcal has been previously analyzed from several points of view. In 2012, Villanueva-Rodríguez and Escalona-Buendía [4] described Mezcal’s elaboration process and different distillation methods. Additionally, Mezcal’s traditional character has also been evaluated [5], and it has also been a case of study to assess product conceptualization based on consumers’ food-related lifestyles [6].

In addition, Vera-Guzman et al. [7] evaluated the effect of agave species, origin and season on minor volatile compound profiles in Mezcals of two local agave varieties in one of the main Protected Designation of Origin (PDO) regions in Oaxaca, Mexico. Notably, Oaxaca is not the only PDO production region in Mexico; Guerrero is also an important region with PDO of craft Mezcal in the country.

Volatiles are distinctive compounds that give distilled alcoholic beverages their unique characteristics. These are affected by many variables, such as the raw materials used, flavor additives and processing steps, which include fermentation, distillation and aging processes [8].

Mezcal has the reputation of being a traditional liquor of good quality, and current regulations allow the use of this title for agave spirits produced throughout the territories of several Mexican regions such as Guerrero. Mezcal producers, which are growing in size and becoming a small industry, have also realized the need for more structured control using sensory methodologies that follow the procedures stated in the International Standards (ISO) as part of their quality management system. However, not all medium or small producers have a proper research and development department; therefore, they often must search for external consultancy [4]. In this vein, organizing and involving mezcal producers in a sensory evaluation of their own products and taking advantage of their product knowledge and expertise could be highly valuable to obtain a broad description of the sensory attributes of mezcal. According to Nimi et al. (2018) [9], expert judges have extensive knowledge and experience with tasting the product and thus can be very useful in the first steps of developing a sensory profile [10]. Moreover, to our knowledge, a sensory characterization tool to assess Mezcal has not yet been developed.

Therefore, the aim of this research was to develop a sensory wheel that includes the attributes that best describe Mezcal from different regions and distillation processes in an objective manner with the assistance of local producers. Additionally, gas chromatography and mass spectrometry were conducted in Mezcal samples to determine the main volatile compounds that confirm the presence of the sensory descriptors. Even though this study focused on Mezcal from the Guerrero area in Mexico, the development of a visual descriptive tool could be helpful as a foundation for differentiating Mezcal from other regions in the country in an easy and practical way.

## 2. Materials and Methods

### 2.1. Experimental Design

This study consisted of creating a sensory lexicon for Mezcal by means of local producers based upon the frequency of their attribute elicitation. Then, a selection of the most relevant descriptors was made to construct a sensory wheel. Subsequently, a sensory panel (eight trained assessors) evaluated a total of 10 Mezcal samples using a reduced list of the sensory lexicon generated by a group of Mezcal producers. Additionally, gas chromatography with mass spectrometry was performed to analyze volatile components’ contribution to the aroma and flavor descriptors.

### 2.2. Sensory Lexicon Development

#### 2.2.1. Mezcal Samples

A total of 20 different Mezcal samples were used for this study. Samples were selected based on production type (mud, cupper and Asian distillers), agave varieties from the Guerrero area (Angustifolia, Americana, Cupreata and Rodacantha) and different agave cultivation zones (north, south, center and east) in the Guerrero state in Mexico (Table 1). Samples were stored at room temperature (25 °C). Mezcal samples were presented in odorless transparent glass cups containing a small amount of Mezcal (around 15 mL), labeled with a random 3-digit code. All 20 samples were used for generating sensory descriptors using the Free Choice Profiling technique (Appendix A).

#### 2.2.2. Free Choice Profiling

Twenty Mezcal producers with at least 10 years of experience with Mezcal from different zones of Guerrero state, Mexico, were recruited to perform Free Choice Profiling [11]. The local producers’ age range was 30 to 75 years old. Producers evaluated 20 different Mezcal samples (Table 1). Producers were not informed about the samples’ origins to avoid biases. Descriptors were generated through a Free Choice Profiling task accomplished in ten tasting sessions. In the first two sessions, producers evaluated the twenty Mezcal samples (ten in each session) to generate the personal attributes they could perceive as relevant when describing each Mezcal sample. In the next eight sessions (five samples per session and a total of two replicates), producers quantified their own elicited attributes and had to rate the same twenty samples in each of them using their own attributes in a lineal scale ranging from 0 to 10, anchored with the words low intensity/absence (0) and high intensity (10). All samples were assessed in a monadic presentation and were presented following a balanced order [12]. Then, validation of the discriminant ability of the selected descriptors between the Mezcal samples for each taster was realized. The descriptors that were discriminant among samples (*p* < 0.25) were consequently retained (Table 2).

#### 2.2.3. Construction of Sensory Wheel

Once Free Choice Profiling was performed, all 20 producers delivered their lists of the elicited sensory terms. The discriminant terms with the same or similar meaning were eliminated, and only the most common terms were used [13]. Only attributes which were relevant, unambiguous, non-redundant and non-hedonic were included [14]. This process was carried out in three sessions in which an open discussion among the 20 producers took place to reach an agreement on the final descriptors to retain.

To simplify the sensory profiling task, the list of preliminary descriptive terms was additionally reduced based on the frequency in which attributes were generated during the Free Choice Profiling sessions (>5%) [14]. All the descriptors of the Mezcal sensory wheel were grouped together to form clear categories based upon an open discussion among participants and the expertise of the researchers involved in the study. The sensory wheel was constructed using the XLSTAT software (Addinsoft, Paris, France) version 2021.1.1.

### 2.3. Descriptive Sensory Analysis of Mezcal

#### 2.3.1. Samples

Ten of the twenty initial samples were used for sensory profiling. Samples were selected to cover different varieties and processing techniques. Thus, two samples of Angustifolia, one sample of Americana, six samples of Cupreata and one sample of Rodacantha agave were included. Samples were presented in transparent plastic cups labeled with a random three-digit code. These samples were served at room temperature (25 °C ± 1 °C), and about 15 mL of each sample was provided for the sensory evaluation. Product samples were presented in a monadic sequence, and two sensory modalities were evaluated: odor and flavor.

#### 2.3.2. Panelists

From the initial sample of 20 local mezcal producers, 8 (3 females and 5 males, aged between 23 and 65 years old) were recruited to perform a Quantitative Analysis of the selected Mezcal samples. These were selected based on their abilities to identify and describe differences in Mezcal (the number of discriminant attributes generated in the Free Choice Profiling process) and their availability to attend the different tasting sessions planned (training and evaluation).

#### 2.3.3. Selection of Sensory Terms for Descriptive Analysis

It has been stated that efficient sensory profiling is possible with 20 attributes [15]. Therefore, the number of attributes was reduced from the total generated by disregarding the individual attributes and focusing exclusively on the families obtained in the sensory wheel (Figure 1).

#### 2.3.4. Sensory Training

Once the final list of descriptors was settled, reference scales were developed for each of the selected sensory descriptors to facilitate panelist training (Table 3). This task was performed over 10 different sessions lasting 4 h each. This procedure was used to help assessors identify what constitutes high and low amounts of each attribute [16]. Flavor and odor scales were developed using specific references. Table 4 shows the characteristics of the different reference scales developed for all attributes and their corresponding score obtained as a result of the consensus between all panelists during the training sessions. Sensory references were set by presenting an array of chemicals, ingredients, spices or products that cover the entire sensory spectrum to be described [17]. Along this process, panelists became familiarized with the different descriptors and their intensity scales to assess the samples in a more accurate form [18].

#### 2.3.5. Descriptive Analysis of Mezcal Samples

Once panelists were familiarized and trained with the sensory descriptors, a quantitative analysis of 10 Mezcal samples was carried out in six different sessions with 3–4 samples each. All the samples were assessed twice (duplicated). In each session, the order of sample presentation and the first-order and carry-over effects were blocked [11]. Sensory evaluation was performed in a test room designed according to ISO guidelines (ISO 8589:2007) [19]. Samples were assessed by means of a semi-structured 10 cm lineal scale anchored in the two extremes (0 = no presence of descriptor; 10 = high intensity) to score all of the selected descriptors. The scoring scale was based on the intensities defined during the training process. All panelists had water to drink and grain crackers to eat as palate cleansers between samples.

### 2.4. Volatile Compound Analysis

Minor volatile compounds were obtained according to the methodology reported in [20]. Before volatile extraction, the samples’ alcohol content was adjusted to 30 mL of ethanol/100 mL by adding distilled water, and ethanol content was verified with calibrated alcoholmeters (Dujardin-Salleron, Paris, France) with the Gay-Lussac scale at 15 °C. Then, 0.2 g of NaCl (Sigma, St. Louis, MO, USA) was added to 325 mL of the alcohol-adjusted (Sigma, St. Louis, MO, USA) sample, and volatiles were extracted with 45 mL of 2-propanol 99% JT Baker. The system was agitated for 5 min and left to stand until complete separation of the organic layer. Finally, extracts were concentrated in a rotary evaporator device at 45 °C until a final volume of 0.6 mL was obtained, which was placed in suitable tightly closed vials and kept at −20 °C until analysis. All the samples were extracted twice.

The obtained extracts were analyzed by gas chromatography in a gas chromatograph HP 5890 Series II coupled to a mass detector (HP 5977AMSD) (Hewlett-Packard, Palo Alto, CA, USA) The compounds were separated in a capillary DB-Wax polyethylene glycol column with 60 m × 0.25 mm ID × 0.25 mm thickness (Hewlett-Packard). The oven was set at 40 °C for 5 min, increasing by 2.5 °C/min until reaching 220 °C and held for 25 min under these conditions. Injector and detector temperatures were kept at 220 °C and 260 °C, respectively. A sample volume of 0.5 mL was automatically injected using helium as carrier gas at 0.7 mL/min, and a 50:1 split ratio was used. The total ion chromatograms (TICs), as well as the mass spectra, were acquired in the electron impact (EI) mode at 70 eV and traced at 1.6 scans/s. Compounds were tentatively identified by comparing the spectrum of each compound with the Nist 14L spectra library MS Interpreter Ver. 3.4.5. Compounds’ identities were confirmed by comparing them with reference standards and/or by comparison with the Kovats index reported in the literature and by consulting bibliographical references of distilled beverages and other spirits whose volatile compositions have been studied in columns like the one used in this study. Quantification was performed with the external standard method with methyl decanoate.

### 2.5. Data Analysis

For FCP data, only discriminant descriptors were retained after performing a two-way ANOVA (Mezcal sample and tasting session) for each assessor (*p* < 0.25).

ANOVA was used to analyze the quantitative data from the trained panel and to identify significant differences between the Mezcal samples. The ANOVA model considered three main effects, namely product (fixed effect), assessor (fixed factor) and session (random effect). The double interaction of product x assessor was also checked and then removed from the final model since it was not significant (*p* > 0.05) in all cases.

For the volatile data, a dissimilarity matrix was first obtained from the values obtained for all samples. Then, multi-dimensional scaling (MDS) was carried out on this dissimilarity matrix to obtain a spatial representation of the most differentiating volatiles among samples. Finally, a Principal Component Analysis (PCA) was performed for the mean values of the odor and flavor data from the sensory analysis and the values from the selected differentiating volatiles. The PCA allowed us to visualize and describe the relationship between the volatiles’ compositions and the odor and flavor descriptors of the Mezcal samples [21]. All statistical analyses were performed using XLSTAT 2018 software (Addinsoft, Paris, France).

## 3. Results

### 3.1. Sensory Wheel

Three hundred and twenty-four terms were collected from Mezcal producers during the Free Choice Profiling sessions. After performing a two-way ANOVA (Mezcal sample and tasting session) for each taster, the list was reduced to 264 terms (Table 2). Subsequently, term selection was performed, and attributes were combined with similar terms until the list was reduced to 194 terms. Of these, 101 attributes were related to odor and 93 to flavor modality.

To create a Mezcal sensory wheel, the number of attributes was further reduced to 87 terms (41 aroma terms and 46 flavor terms) based on the frequency of quotation. The attributes were assembled in a three-tiered wheel (Figure 1). The descriptors forming the outer tier are specific attributes such as “lemon”, “marigold”, “fresh herbs”, etc. The secondary tier comprises ten primary descriptors associated with odor (alcoholic, citric, earthy, floral, herbal, maguey mentholated, smoked, spice and wood) and thirteen more for flavor (alcoholic, astringent, bitter, citric, earthy, floral, fresh, fruity, maguey, mineral salt, smoked, spice and wood). These attributes are generic terms that group together similar descriptors found in the outer tier. The inner tier contains two major sensory modalities, namely odor and flavor.

### 3.2. Descriptive Analysis

All secondary tier descriptors (10 for odor modality and 13 for flavor) were selected for the final descriptive profiles of the 10 selected samples (Figure 1).

Table 5 shows the results of the quantitative analysis. The sample of the Cupreata variety made with earthenware distillation (560) showed the highest value for citric, earthy and alcoholic odors and was one of the samples with the highest value for a citric flavor. The sample of the Cupreata variety from a mountain area made with cupper distillation (344) exhibited the highest values of herbal and flavor odors and the highest values for fruity and maguey flavors. The sample of the Angustifolia variety from a jungle area made with cupper distillation (387) showed the highest floral odor and the highest value for an astringent flavor, while the sample of the Americana variety from a mountain area made with cupper distillation (399) had the highest mentholated and spicy odors and the highest values for bitter, fresh and smoked flavors; it was also one of the samples with the highest value for citric flavor. The sample of the Cupreata variety from forest land made with cupper distillation (613) had the highest value for a smoked odor, and the sample of the Cupreata variety from humid land made with cupper distillation (327) had the highest value for wood odor and the highest values for floral, fruity and mineral salt flavors. The sample of the Angustifolia variety from a mountain area made with cupper distillation (600) had the highest mean values for earthy and spicy flavors, and the sample of the Cupreata variety from a humid zone made with cupper distillation (512) had the highest value for wood flavor. Finally, the sample of the Cupreata variety made with clay soil and cupper distillation (428) had the highest value for alcoholic flavor.

### 3.3. Volatile Compound Identification

In this work, 111 minor volatile compounds were identified by means of mass spectrum and retention indexes reported in the literature [20]. Table 6 shows the frequency in which these compounds were found from the total assessed samples and the literature descriptors associated with them.

The highest number of identified compounds were esters (37), which are generally associated with a fruity odor [22]. The second largest group was alcohols (17). These were present in all the Mezcal samples, and they mainly originate from lipid oxidation and reduction reactions of aldehydes.

The next group of identified compounds were ketals and terpenes (12 and 9 compounds, respectively). Of the total identified compounds, 54 were found in all the samples, 8 in 90% of the samples, 14 in 80%, 8 in 70% of samples, 3 in 60%, 4 in 50%, 4 in 40% and 4 in 30% of the samples. The compounds with lower numbers of appearances were alkanes and phenols (three and five) (Table 6).

### 3.4. Principal Component Analysis

PCA plots were used to display the overall differences between samples. Figure 2 shows the biplot for odor (a) and flavor (b) descriptors and for the relationship with the selected (differentiating) volatile compounds (*n* = 25).

Figure 2a shows that the Rhodacantha, Angustifolia and Americana varieties were located closer to each other on one side of the plot, whereas several Cupreata samples were positioned in the opposite side of the plot. Cupreata Mezcal was associated with alcoholic, earthy, citric, smoky and woody odors. The odor attributes floral, fruity, herbal, mentholated and spicy were associated with the other three varieties. Half of the differentiating volatile compounds were associated with different samples of the Cupreata variety, and the rest of them to the other three Mezcal varieties.

The visual relationship between Mezcal flavor descriptors and volatiles is presented in Figure 2b. It can be observed that most flavor attributes were associated with Mezcals of the Cupreata variety. In relation to the volatile compounds, most of them were associated with different samples of the Cupreata variety.

Both volatiles and sensory descriptors in Figure 2a,b that are in green are those that are best represented in the two dimensions of the biplot (squared cosines of the variables). The sensory descriptors in blue as well as the volatile compounds in black have a lower representation in the two dimensions of the biplot.

Sample 399 of Mezcal of the Americana variety had mentholated and spicy odors and included the presence of anethol and verbenone, which are both compounds associated with spicy odors. The spice odor comes from the fermentation process in wood in which aromas from oak can be transferred to the maguey [7]. Samples 387 and 600, both of the Angustifolia variety and made with cupper distillation, had the greatest floral odor and the largest presence of iso-pentanol, phenyl-ethyl alcohol, thus confirming the link between the odor and presence of volatiles. Floral compounds such as phenyl-ethyl alcohol have previously been reported in spirits made from different agave species [7]. Samples of the Rhodacantha variety made with Asian distillation were associated with an herbal odor and included the presence of a hexanol compound that has been related to green odors.

Sample 560 of the Cupreata variety made with earthenware distillation had the highest alcohol odor and was also associated with the presence of methyl-pentanol, which is a well-known type of alcohol aroma according to the literature. Sample 344 of the Cupreata variety from a mountain region made with cupper distillation had distinctive herbal and fruity odors confirmed by the presence of ethyl acetate, which is a known fruit aroma compound. Additionally, this agave variety was also associated with a maguey flavor. The presence of compounds such as furfural and derivatives could be produced by a Maillard reaction during agave cooking [23]; another study quantified furfural-type compounds at different cooking times for *A. tequilana* Weber and attributed their generation to Maillard reactions [7].

The rest of the samples of the Cupreata variety made with cupper distillation (327, 613, 512 and 428), were associated with woody, citric and smoky odors. Volatile phenolic derivatives have been associated with smoked foods and certain characteristics of alcoholic beverages [22]. Phenol and phenol derivatives may be formed by the thermal degradation of lignin during the cooking process to produce mezcal [23]. One clear phenol presence among samples was o-Ethyl phenol.

## 4. Discussion

The aim of this study was to generate an extensive list of sensory attributes and develop a sensory wheel that facilitates Mezcal description with a Protected Designation of Origin. This description should enable a differentiation between different types of Mezcal according to their raw materials (maguey), processing techniques and places of origin. It is crucial for producers to know which sensory descriptors are relevant when characterizing their products [24] and how the raw material and the applied process impact the sensory properties of the final product. Thus, the sensory wheel developed in this study can be used both as a quality control tool and as a marketing tool that allows producers to differentiate their products in the market. Since Mezcal sensory description in previous studies has been performed with a descriptive analysis and only by means of laboratory-controlled evaluation assessors, including the perspectives of producers should cover a more market-oriented approach. Therefore, this study provides an applicable instrument for Mezcal sensory evaluation from a different and local perspective, which gives practical meaning to Mezcal producers.

Concerning sensory description, all odor and flavor attributes showed significant differences (*p* < 0.05), thus showing that the attributes included in the sensory wheel are appropriate to differentiate the different types of Mezcal samples (based on agave varieties, processes and regions).

One example of the sensory attributes’ discrimination capacity was observed between agave varieties. Mezcal of the Angustifolia variety was characterized with the floral odor attribute. This variety has been cultivated in a forest-type region where different wild flowers grow around the land where this agave is cultivated, thus having an intrinsic floral odor that clearly remains through Mezcal elaboration.

Regarding differences in processes, distillation materials in Mezcal play an important role since the product is in contact with these materials for several hours. Within this context, sample 560 of the Cupreata variety made with an earthenware distillation process was characterized with the earthy odor attribute compared to the other samples. In this vein, it has been stated that in a distillation process where a clay pot is utilized, muddy odors can be described [4]. On the other hand, the samples the same variety (Cupreata) made with a cupper distillation process was characterized with the smoky and woody odor attributes, which can be explained by this sample’s process of agave cooking with burning wood and a utilized metal recipient for distillation. Therefore, the aforementioned attributes can be useful for differentiating Mezcal processes.

On the other hand, the quality and authenticity of Mezcal are highly relevant issues because of the beverage’s unique alcoholic flavor and odor, which are the result of the agave type and the volatile compounds [7]. As was observed in this study, different volatile compounds characterize different samples depending on the distillation process, raw material (agave species) and fermentation conditions. Additionally, most of the volatile compounds that were found in GC-MS could be related to the primary attributes from the sensory aroma and flavor wheel, thus confirming the presence of different sensory attributes. A clear example was the presence of esters, which were abundantly present in some Mezcal samples. These compounds have also been proven to be responsible for a fruity aroma and flavor in many distilled beverages [25].

Consequently, it could be said that the odor and flavor attributes selected for the sensory wheel were accurate for Mezcal description. Additionally, the trained panel was able to distinguish the different types of Mezcal using the sensory lexicon developed by producers, which means the training process was effective and the developed references were accurate to define attribute intensities. Since Mezcal is a PDO product, it is necessary to harmonize the definition of attributes and sensory references that allow for its characterization [26]. Moreover, because a standardized method of assessing Mezcal has not been established before, the appropriate sensory references developed in this study could also be useful for training future panelists from other PDO regions [27].

After training the assessors on the aforementioned attributes, the performance of the panel in the QA proved that they were capable of differentiating the samples depending on the agave species used for Mezcal production, their origin and even the type of distillation process used (cupper or earthenware distillation).

## 5. Conclusions

The created sensory profile (sensory wheel) could be a valuable instrument for the Mezcal industry. It can be used for comparing and monitoring product quality and characteristics associated with a production process and/or a product from a specific place of origin. This instrument can allow producers to not only evaluate their product but also be able to differentiate other types of Mezcal from their own. The development of the specific sensory references in this work proved to be effective for assessor’s repeatability. Additionally, the fact that there was a panel consensus on the attributes used for the different types of samples demonstrated the accuracy of these references despite Mezcal’s difficult nature regarding characterization (high alcohol content).

The descriptors used for this wheel are limited to the 20 Mezcal samples that were analyzed from two seasons of the production of agave from the Guerrero state area. Thus, it can be expected that certain modifications should be made over time as the wheel is presented to consumers and used by the Mezcal industry. Additional research will be necessary to improve the wheel or to include other Protected Designation of Origin Mezcal wheels different from that of the Guerrero state. Also, by analyzing a more considerable number of Mezcal samples from different harvest seasons and different cities with a PDO, the Mezcal odor and flavor wheel can be expanded and validated even further. Nevertheless, this study can be a basis for the sensory analysis of Mezcals with different agave varieties, distillation processes and origins.

## Figures and Tables

**Figure 1 foods-14-00402-f001:**
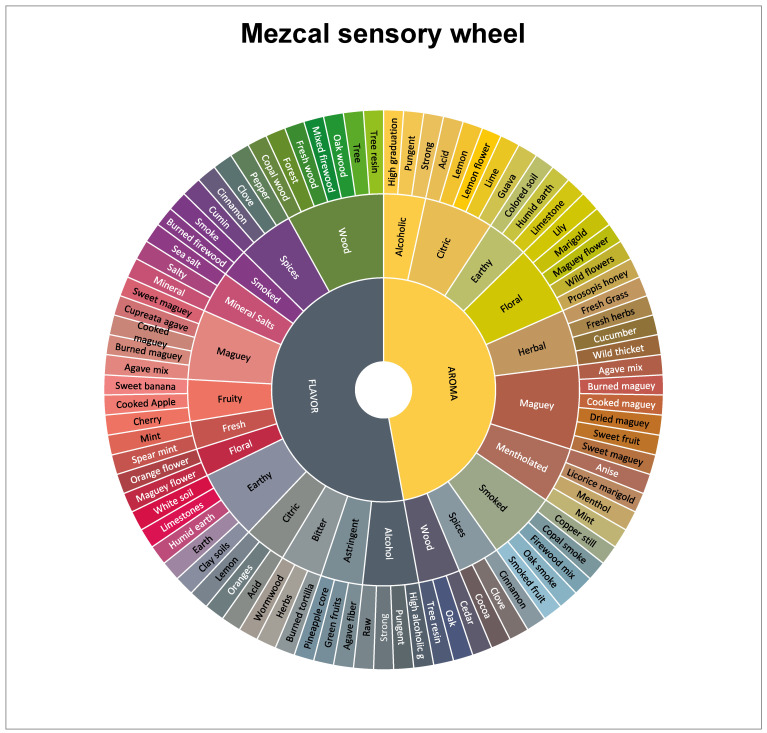
Mezcal sensory wheel.

**Figure 2 foods-14-00402-f002:**
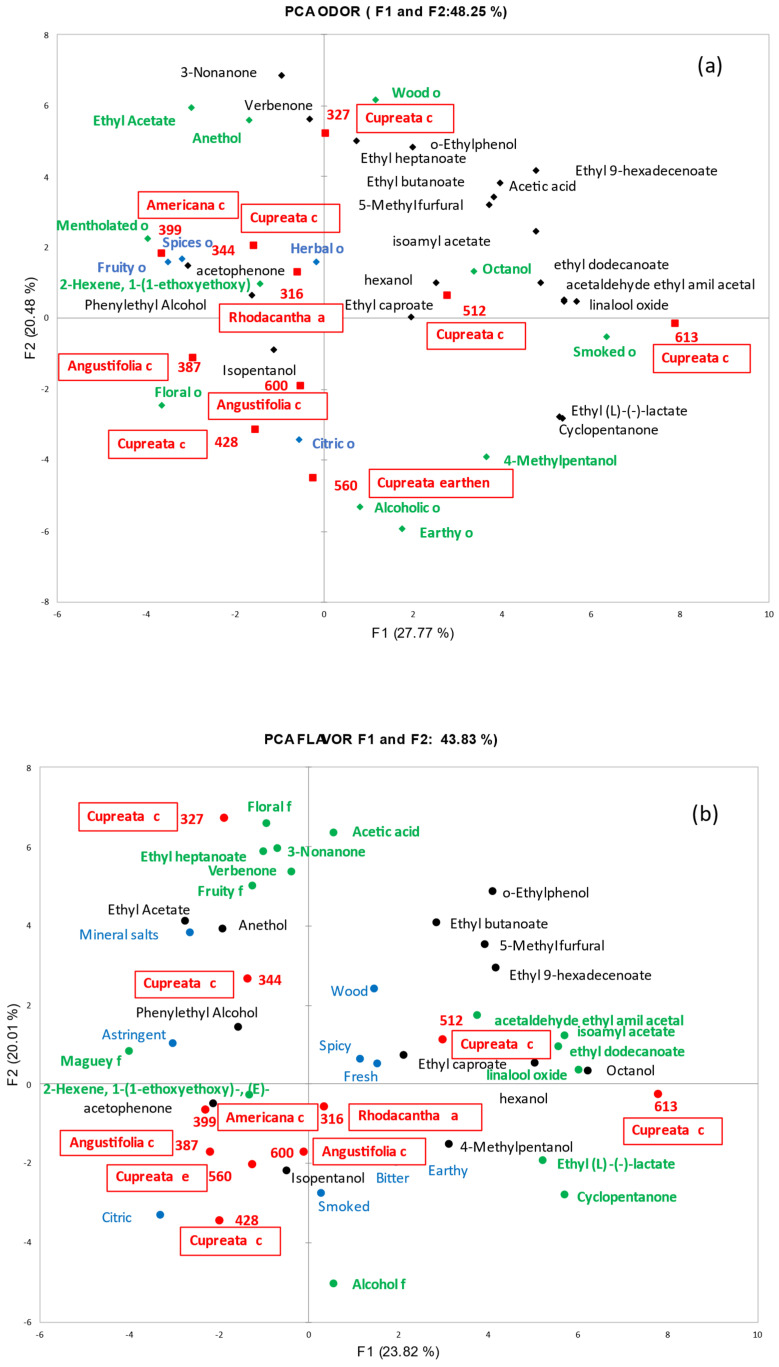
Principal Component Analysis results of odor (**a**) and flavor (**b**). Letters in red squares next to the sample numbers indicate sample names. (Letter c indicates cupper distillation, letter e refers to earthen distillation and letter a refers to Asian distillation). Letters in green indicate sensory descriptors (f for flavor and, o for odor) and volatiles associated to the samples that are represented in the first two dimensions of the biplot. Letters in blue indicate the sensory descriptors associated to the samples not represented in the first two dimensions of the biplot. Letters in black indicate volatile compounds not represented in the first two dimensions of the biplot.

**Table 1 foods-14-00402-t001:** Description of Mezcal samples for lexicon development.

Agave Variety	Distillation Type	Location in Mexico	Sample Code
Angustifolia	Cupper distiller	Ahuacuotzingo Trapiche	600 *
Angustifolia	Cupper distiller	Ahuacuotzingo Motuapa	
Angustifolia	Cupper distiller	Huitzuco Atetela	387 *
Angustifolia	Cupper distiller	Huitzuco Paso Morelos	
Angustifolia	Cupper distiller	Yerbabuena	
Americana	Cupper distiller	Centro Montaña	399 *
Americana	Cupper distiller	Chilpancingo	
Cupreata	Cupper distiller	Zihuaquio	512 *
Cupreata	Cupper distiller	Chilapa Santa Cruz	428 *
Cupreata	Cupper distiller	Chilapa Ayahualco	
Cupreata	Cupper distiller	Chilapa Los Amates	
Cupreata	Cupper distiller	Ahuihuiyuco	327 *
Cupreata	Cupper distiller	Amojijeca	613 *
Cupreata	Cupper distiller	Acatlán	344 *
Cupreata	Earthenware distiller	Sierra	560 *
Cupreata	Cupper distiller	Atenango Coacan	
Cupreata	Cupper distiller	Zitala	
Cupreata	Earthenware distiller	Martir de Cuilapan	
Rhodacantha	Cupper distiller	Zihuaquio	
Rhodacantha	Asian distiller	Zihuaquio	316 *

* Samples used for QA and chemical analysis.

**Table 2 foods-14-00402-t002:** Total elicited and significant attributes from Free Choice Profiling.

Taster	Generated Odor Attributes	Generated Flavor Attributes	Discriminant OdorAttributes, *p* < 0.25	Discriminant Flavor Attributes, *p* < 0.25
1	5	9	5	8
2	7	8	6	7
3	6	10	4	9
4	6	9	5	8
5	6	9	4	7
6	7	11	6	9
7	6	9	5	8
8	6	9	5	8
9	6	9	4	8
10	8	11	5	9
11	9	10	7	9
12	7	9	6	8
13	6	10	5	8
14	6	13	4	10
15	7	8	4	7
16	6	8	4	7
17	8	9	6	8
18	7	10	5	9
19	8	10	6	7
20	7	9	5	8
Total	134	190	101	163
Total final		324		264

**Table 3 foods-14-00402-t003:** Selected descriptors for the QA.

Attributes	Description
Odor	
Alcohol	Pungency derived from present alcohol intensity
Citric	Odor intensity of citric fruits like lemon
Earthy	Odor intensity like humid earth
Herbal	Intensity of herbal odor like fresh cut grass
Floral	Odor intensity of flowers
Maguey	Odor intensity like cooked maguey (agave)
Menthol herbs	Odor intensity like mint
Smoked	Odor intensity like firewood
Spices	Odor intensity like cinnamon or clove
Wood	Odor intensity like cedar wood
Flavor	
Alcohol	Flavor like high alcohol intensity
Astringent	Flavor like pineapple core
Bitter	Flavor like quinine
Citric	Flavor like orange or tangerine
Earthy	Flavor like humid earth
Floral	Flavor like flowers
Fresh	Flavor like mint
Fruity	Flavor like apple or banana
Maguey	Flavor like cooked agave
Mineral Salts	Flavor like mineral salts
Smoked	Flavor like smoke from firewood
Spices	Flavor like cumin clove
Wood	Flavor like fresh wood

**Table 4 foods-14-00402-t004:** Odor and flavor references for training.

Sensory Modality	Descriptor	Product Used	% of Product Used	Part of the Scale	Score
Odor	Alcohol	Heads of Mezcal distillation 75° GL	50	High	8–10
			40	Medium	5–7
			30	Low	1–3
	Citric	Orange and tangerine peel, 5% solution (boiled in water)	50	High	8–10
			15	Medium	5
			2.5	Low	2
	Earthy	Earth solution (400 mg in 100 mL)	45	High	8–10
			35	Medium	5
			25	Low	1–2
	Herbal	2.5% solution of painted cup herbs (*Castilleja tenuiflora*)	10	High	8–10
			2.5	Medium	5
			1.25	Low	1–2
	Floral	15% solution of dry flowers (*Tagetes erecta*)	3.84	High	8–10
			1.96	Medium	5
			0.4	Low	1–3
	Maguey	Cooked *Cupreata agave* solution 50%	60	High	8–10
			33	Medium	5
			10	Low	1–2
	Menthol Herbs	15% solution of mint tea Alessa gourmet tea, 5% solution of Tagetes micrantha Cav Mexico	100	High	8–10
			60	Medium	5
			15	Low	1–2
	Smoked	Cooked maguey with wood (72 h)	100	High	8–10
			66	Medium	5
			40	Low	1–3
	Spices	0.13% pepper, 0.5% clove, 0.5% oregano solution	100	High	8–10
			50	Medium	5
			15	Low	1–3
	Wood	50% solution (cedar, oak and pine in alcohol) of distilled extract	100	High	8–10
			30	Medium	5
			10	Low	1–3
Flavor	Alcohol	Heads of Mezcal distillation 75° GL	50	High	8–10
			40	Medium	5
			30	Low	1–3
	Astringent	10% solution of pomegranate skin (*Punica granatum*) white interior	75	High	8–10
			50	Medium	5
			15	Low	1–3
	Bitter	0.05% solution of painted cup herbs (*Castilleja tenuiflora*)	2.27	High	8–10
			1.44	Medium	5
			0.66	Low	1–3
	Citric	2.5% solution of passion fruit (*Passiflora edulis*)	3.84	High	8–10
			1.96	Medium	5
			0.4	Low	1–3
	Earthy	Earth solution (400 mg in 100 mL)	45	High	8–10
			35	Medium	5
			25	Low	1–2
	Floral	40% solution of dry flowers (*Tagetes erecta*)	3.84	High	8–10
			1.96	Medium	5
			0.4	Low	1–3
	Fresh	15% solution of mint tea Alessa gourmet tea Mexico	100	High	8–10
			60	Medium	5
			15	Low	1–2
	Fruity	10% solution of apple, banana and pear porridge	100	High	8–10
			50	Medium	5
			25	Low	1–3
	Maguey	Cooked *Cupreata agave* solution 50%	60	High	8–10
			33	Medium	5
			10	Low	1–3
	Mineral Salts	1.5% NaCl solution	15	High	8–10
			7.5	Medium	5
			2.5	Low	1–3
	Smoked	Cooked maguey with wood (72 h)	100	High	8–10
			66	Medium	5
			40	Low	1–3
	Spices	0.13% pepper, 0.5% clove, 0.5% oregano solution	100	High	8–10
			50	Medium	5
			15	Low	1–3
	Wood	50% solution (cedar, oak and pine in alcohol) of distilled extract	100	High	8–10
			30	Medium	5
			10	Low	1–3

**Table 5 foods-14-00402-t005:** Mean values of odors and flavors in QA.

Descriptor	399	316	344	327	600	512	613	428	387	560
Odor										
Alcohol	4.88 ^bc^	5.00 ^bc^	4.88 ^bc^	4.25 ^bc^	4.13 ^c^	5.13 ^bc^	5.50 ^bc^	5.62 ^ab^	5.38 ^bc^	7.00 ^a^
Citric	3.13 ^ab^	2.44 ^ab^	2.62 ^ab^	1.56 ^bc^	2.56 ^ab^	1.63 ^bc^	2.25 ^abc^	0.44 ^c^	2.63 ^ab^	3.56 ^a^
Earthy	0.00 ^b^	0.06 ^b^	0.00 ^b^	0.00 ^b^	0.63 ^b^	0.13 ^b^	0.68 ^b^	0.50 ^b^	0.06 ^b^	1.68 ^a^
Herbal	1.50 ^bc^	2.62 ^ab^	3.44 ^a^	0.75 ^bc^	1.94 ^abc^	1.81 ^bc^	1.38 ^bc^	0.75 ^c^	0.75 ^c^	1.38 ^bc^
Floral	0.38 ^bc^	0.25 ^bc^	1.06 ^bc^	0.31 ^bc^	0.63 ^bc^	0.44 ^bc^	0.00 ^c^	1.18 ^ab^	2.25 ^a^	0.63 ^bc^
Fruity	2.81 ^bc^	3.25 ^ab^	4.63 ^a^	1.75 ^bc^	2.00 ^bc^	2.25 ^bc^	1.63 ^bc^	1.56 ^c^	3.25 ^ab^	2.56 ^bc^
Menthol herbs	2.87 ^a^	1.75 ^abc^	2.19 ^ab^	1.19 ^bcd^	0.56 ^d^	1.31 ^bcd^	0.81 ^cd^	0.88 ^cd^	2.31 ^ab^	1.50 ^bcd^
Smoked	0.19 ^d^	1.25 ^bcd^	1.06 ^bcd^	2.00 ^abc^	1.56 ^bcd^	2.56 ^ab^	3.56 ^a^	1.44 ^bcd^	0.63 ^cd^	1.94 ^abc^
Spices	4.88 ^a^	1.50 ^b^	1.00 ^b^	1.00 ^b^	1.50 ^b^	0.88 ^b^	0.50 ^b^	1.75 ^b^	1.00 ^b^	0.50 ^b^
Wood	1.38 ^abc^	1.00 ^bc^	1.44 ^abc^	2.63 ^a^	1.25 ^bc^	1.88 ^ab^	1.31 ^bc^	1.25 ^bc^	0.81 ^bc^	0.31 ^c^
Flavor										
Alcohol	6.25 ^ab^	5.63 ^bc^	5.75 ^abc^	4.50 ^c^	5.50 ^bc^	6.63 ^ab^	6.13 ^ab^	7.19 ^a^	6.75 ^ab^	5.63 ^bc^
Astringent	4.63 ^ab^	3.75 ^abc^	3.13 ^bc^	4.68 ^ab^	2.56 ^c^	2.13 ^c^	3.31 ^bc^	3.63 ^bc^	5.38 ^a^	2.88 ^c^
Bitter	5.44 ^a^	3.44 ^abcd^	2.13 ^d^	2.00 ^d^	2.25 ^d^	3.38 ^bcd^	4.63 ^ab^	2.25 ^d^	4.50 ^abc^	2.50 ^cd^
Citric	2.31 ^a^	1.38 ^ab^	1.75 ^ab^	0.75 ^b^	1.13 ^ab^	0.63 ^b^	1.06 ^ab^	1.50 ^ab^	1.63 ^ab^	2.19 ^a^
Earthy	0.00 ^b^	0.00 ^b^	0.00 ^b^	0.00 ^b^	0.75 ^a^	0.00 ^b^	0.38 ^ab^	0.00 ^b^	0.13 ^b^	0.00 ^b^
Floral	0.50 ^ab^	0.38 ^b^	1.00 ^ab^	1.50 ^a^	0.50 ^ab^	1.00 ^ab^	0.25 ^b^	0.38 ^b^	0.25 ^b^	0.13 ^b^
Fresh	3.31 ^a^	2.31 ^abc^	1.75 ^bc^	2.25 ^abc^	2.25 ^abc^	2.00 ^abc^	2.62 ^ab^	1.75 ^bc^	0.94 ^c^	2.44 ^ab^
Fruity	1.25 ^c^	2.25 ^c^	4.94 ^a^	4.31 ^ab^	3.00 ^bc^	2.63 ^bc^	1.75 ^c^	2.19 ^c^	1.31 ^c^	2.75 ^bc^
Maguey	3.93 ^b^	3.38 ^b^	5.94 ^a^	3.75 ^b^	3.88 ^b^	3.75 ^b^	2.68 ^b^	4.00 ^b^	4.25 ^ab^	4.56 ^ab^
Mineral salt	0.69 ^abc^	0.06 ^c^	0.75 ^abc^	1.13 ^a^	0.19 ^bc^	0.44 ^abc^	0.38 ^abc^	0.38 ^abc^	0.63 ^abc^	1.00 ^ab^
Smoked	4.31 ^a^	2.56 ^bcd^	0.94 ^e^	1.75 ^de^	1.69 ^de^	3.63 ^ab^	2.38 ^bcd^	3.38 ^abc^	1.81 ^cde^	2.81 ^abc^
Spices	0.69 ^c^	1.00 ^c^	1.94 ^ab^	0.88 ^c^	2.50 ^a^	1.00 ^c^	1.25 ^bc^	0.56 ^c^	0.75 ^c^	0.88 ^c^
Wood	1.81 ^bc^	1.18 ^c^	2.88 ^ab^	1.81 ^bc^	1.75 ^bc^	3.94 ^a^	1.75 ^bc^	1.38 ^c^	1.81 ^bc^	1.88 ^bc^

Different letters in the same row mean significant differences, *p* < 0.05.

**Table 6 foods-14-00402-t006:** Volatile compounds detected via GC-MS.

Compound	D.F.	Identification	Descriptors in the Literature
ALCOHOLS			
2-Ethyl-1-hexanol	80	MS RI	Honey
Butanol	80	RI	Wine, fermented
Isopentanol	70	MS	Vanilla
Hexanol	80	MS	Green/earthy
4-Methylpentanol	30	MS	fermented
3-Octanol	50	MS	Herbs
5-MethyIfurfural	100	MS	Amaretto
Octanol	100	MS	Green
1-Decanol	50	MS	Fat
1-Phenyl-2-propanol	40	MS	Floral
Benzyl alcohol	70	MS	Jazmin
Phenylethyl Alcohol	90	MS	Rose
1-Dodecanol	90	MS	Coconut oil
Phenol, 2-ethyl-4-methyl-	100	MS	Spicy
1-Tetradecanol	100	MS	Wax
p-Ethylphenol	100	MS	Woody
1-Hexadecanol	100	MS	Fatty
ACIDS			
Acetic acid	80	MS RI	Sour vinegar
Isobutyric acid	90	MS RI	Sharp butter
Butanoic acid	100	MS RI	Wine, oily
Isovaleric acid	100	MS RI	Fruity
Octanoic acid	60	MS RI	Wine
Decanoic acid	100	MS RI	Earthy
Hexadecanoic acid	100	MS RI	Wax
ALKANES			
Verbenone	100	MS	Rosemary
Anethole	80	MS	Anise spicy
Cyclopentanone, 2-ethyl-	100	MS	Mint
ALDEHYDES			
Isovaleraldehyde, diethyl acetal	100	MS	Sweet fruit
Acetaldehyde ethyl amyl acetal	90	MS	Alcohol
Benzaldehyde	80	MS	Bitter, almond
Phenylacetaldehyde diethyl acetal	40	MS	Green
Cinnamaldehyde	100	MS	Cinnamon
3-Ethoxypropionaldehyde diethyl acetal	50	MS	Pungent
KETONES			
Cyclopentanone	80	MS	Peppermint
3-Ethylcyclopentanone	80	MS	Grassy, musty
3-Nonanone	90	MS	Floral
2-Nonanone	90	MS	Rose tea
Acetophenone	60	MS	Acacia flower, musty
(−)-Car-3-en-2-one (timol)	100	MS	
Phenylacetone	30	MS	Almond
3-Methylacetophenone	90	MS	Acacia
KETALS		MS	
Butane, 1-(1-ethoxyethoxy)-	90	MS	Wine
2-Hexene, 1-(1-ethoxyethoxy)-, (E)-	70	MS	Green
Butiraldehyde diethyl acetal	30	MS	
p-Menth-2-en-7-ol, trans- 2- Cyclohexene-1-methanol, 4-1 methylethyl	100	MS	
Naphthalene, 1,7-dimethyl-	90	MS	Pungent
γ-Nonalactone	70	MS	Sweet milk
Benzene, (2,4-cyclopentadien-1-ylidenemethyl)-	90	MS	Alcohol
Diisooctyl phthalate	50	MS	Solvent
Dibutyl phthalate	100	MS	Oily
Anthracene	100	MS	
Phtalate	90	MS	Solvent
Furfuryl ether	80	MS	Spicy
ESTERS			
Ethyl butanoate	70	MS	Pineapple
Ethyl succinate	100	MS	Cooked apple
Citronellyl butyrate	70	MS	Fruity, raspberry
Ethyl phenacetate	90	MS	Floral
Phenethyl acetate	60	MS	Sweet rose
Ethyl dodecanoate	100	MS	Waxy
Isopentyl decanoate	100	MS	
Ethyl 3-phenylpropanoate	100	MS	
Butanedioic acid, ethyl 3-methylbutyl ester	80	MS	
Ethyl Acetate	30	MS	Fruity anise
Ethyl propanoate	70	MS	
Ethyl butyrate	40	MS	Fruity
Isobutyl acetate	100	MS	Fruity
Isoamyl acetate	100	MS	Banana
Ethyl caproate	30	MS	
Ethyl heptanoate	100	MS	Fruity
Ethyl (L)-(−)-lactate	100	MS	Buttery
Ethyl 2-hydroxybutyrate	100	MS	Fruity
Ethyl 2-hydroxyisovalerate	100	MS	Fruity
Ethyl octanoate	100	MS	Wine, brandy, fruity
Ethyl nonanoate	100	MS	Fruity
Isoamyl lactate	100	MS	
Ethyl levulinate	100	MS	
Ethyl decanoate	30	MS	Fruity
Phenylpropyl acetate	90	MS	
Ethyl (Z)-cinnamate	100	MS	
Ethyl tetradecanoate	100	MS	
Ethyl pentadecanoate	60	MS	
Ethyl hexadecanoate	80	MS	
Ethyl 9-hexadecenoate	100	MS	Fruity
Methyl oleate	100	MS	
Ethyl Oleate	80	MS	
Ethyl linoleate	80	MS	
Ethyl-9,12-octadecadienoate	50	MS	
Methyl linolenate	70	MS	
Ethyl linolenate	80	MS	
Benzyl Benzoate	100	MS	Fruity
PHENOLS			
Dihydrochavicol	30	MS	Herbal
o-Cresol	100	MS RI	Smoke
3,5-Di-tert-butylphenol	100	MS	Burnt sugar
o-Ethylphenol	100	MS	Woody/smoky
Thymol	100	MS	Spicy
FURANS			
Methyl-4,5-dihydro-3(2H)-furanone	100	MS RI	Buttery, coconut
2-Furaldehyde diethyl acetal	100	MS RI	Almonds
Furfural	100	MS RI	Caramel
2-Acetylfuran	100	MS RI	Balsamic
5-MethyIfurfural	100	MS RI	Caramel
Benzofuran, 2-methyl-	100	MS RI	Burnt
Ethyl 3-furoate	100	MS RI	Burnt smoky
TERPENES			
β-Ocimene	100	MS RI	Floral, woody
Terpinen-4-ol	80	MS RI	Citrus
α-Terpineol	40	MS RI	Earthy
cis-Geraniol	100	MS RI	Floral rose citric
Nerolidol, E-	100	MS RI	Firewood
Bisabolol oxide	70	MS RI	Flower
δ-Cadinol	100	MS RI	
Farnesol	100	MS RI	Violet
Linalool oxide	100	MS RI	Flower

D.F.: Detection frequency in samples. MS: Compounds identified based on the IE mass spectra in the Nist 14 library. RI: Compounds identified on the basis of retention index from the literature.

## Data Availability

The original contributions presented in this study are included in the article. Further inquiries can be directed to the corresponding author.

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
