# Peer review of "Mezcal Characterization Through Sensory and Volatile Analyses"

_foods, 2025, doi:10.3390/foods14030402_

Round 1
Reviewer 1 Report
Comments and Suggestions for Authors
The authors have generated comments describing the sensory attributes based on the differences in distillation techniques and producing regions of the liquor, and has formed a standardized flavor wheel. This work holds significant practical value for the protection of geographical indications of products.
This is an important research paper on beverage.
Some suggestions: Currently, artificial intelligence, electronic nose, and electronic tongue technologies are highly developed. Has the author considered utilizing these technologies to conduct objective evaluations of the products? The author may add this in the discussion section.
The sensory attribute descriptions of other types of liquor (such as Chinese Baijiu) are relatively mature. The author can conduct comparative discussions on this aspect.
Regarding other textual editing errors:
Line 27, “It´s consumption” should be “Its consumption”
Line 65, “a good quality traditional spirit,” Rewrite this sentence
Line 145, “15ml” should be “15 ml”; Line 193, “0.6mL”
Line 213, “a two- way”
Reviewer 2 Report
Comments and Suggestions for Authors
Lazo et al. have aimed to develop a sensory wheel for Mezcal from different regions and distillation processes and determine volatile composition to confirm the presence of sensory descriptors. Although the work reported in this article is of interest in the field, the following points need to be addressed for a possible publication in Foods:
1. The title in the submission site is different from that in the manuscript file.
2. The abstract is written with a list of the theoretical methodologies used in the study. But it fails to provide the exact findings in terms of optimized quantitative data on sensory and volatile analyses. So the abstract needs to be rewritten with specific elucidations from this study in terms of results.
3. The keywords are simple and do not completely capture the study. The authors should include some more keywords such as ‘mezcal sensory wheel’, ‘sensory panel training’, ‘Mexico mezcal’, ‘volatile analysis’, ‘distillation process’, ‘aroma and flavor’.
4. All the purchase details of chemicals/reagents and instruments/equipment/software/kits should be provided as state, city, and country in the case of USA as well as city and country in the case of other countries. Also, the authors can just mention the company name for the second instance.
5. The version details of all software used should be specified.
6. The 2nd column of Table 1 is incomplete with the information.
7. The significance letters on each data value should be superscripted for data clarity.
8. Figure 2A and 2B should be enlarged and place one below the other for clarity of labels.
9. A copy of the survey questions or questionnaires collected from panelists should be provided in the supplementary material and cited in the main text. If it is of different language, then it must be translated and included.
10. The number of panelists (8) seems to be less when optimizing these many descriptors for the development of standardized sensory wheel.
11. Why the quantitate data of volatile compounds were not provided? It is mentioned in section 2.4 that they were quantified.
12. Why the consumer validation on the developed sensory wheel was not done? It could enhance the practical relevance for broader market applications.
13. It should be better to make a generalized sensory evaluation for Mezcal that can not only be applied to samples from specific regions, but also from other regions and producers.
14. It should be better to investigate how seasonal variations in agave harvesting and production processes influence sensory profiles and volatile composition.
15. The conclusion should contain only one or two paragraphs, probably with the first paragraph summarizing the key findings, while the second paragraph highlighting the limitation of this study and future perspectives.
Round 2
Reviewer 2 Report
Comments and Suggestions for Authors
The authors have satisfactorily addressed all the comments raised by reviewers and substantially improved the overall quality of the article. Therefore, I recommend accepting this article for publication in Foods.